# Histone Abundance Quantification via Flow Cytometry of Htb2-GFP Allows Easy Monitoring of Cell Cycle Perturbations in Living Yeast Cells, Comparable to Standard DNA Staining

**DOI:** 10.3390/jof9101033

**Published:** 2023-10-20

**Authors:** Maria V. Kulakova, Eslam S. M. O. Ghazy, Fedor Ryabov, Yaroslav M. Stanishevskiy, Michael O. Agaphonov, Alexander I. Alexandrov

**Affiliations:** 1Federal Research Center of Biotechnology of the RAS, Bach Institute of Biochemistry, Leninskiy Ave. 33, Moscow 119071, Russia; 2Institute of Biochemical Technology and Nanotechnology, Peoples’ Friendship University of Russia (RUDN), 6 Miklukho-Maklaya Street, Moscow 117198, Russia; stanishevskiy-yam@rudn.ru; 3Department of Microbiology, Faculty of Pharmacy, Tanta University, Tanta 31111, Egypt; 4Weizmann Institute of Science, Herzl Str. 234, Rehovot 7610001, Israel

**Keywords:** yeast, cell cycle, histone, *Ogataea*, *Saccharomyces*, flow cytometry

## Abstract

Assaying changes in the amount of DNA in single cells is a well-established method for studying the effects of various perturbations on the cell cycle. A drawback of this method is the need for a fixation procedure that does not allow for in vivo study nor simultaneous monitoring of additional parameters such as fluorescence of tagged proteins or genetically encoded indicators. In this work, we report on a method of Histone Abundance Quantification (HAQ) of live yeast harboring a GFP-tagged histone, Htb2. We show that it provides data highly congruent with DNA levels, both in *Saccharomyces cerevisiae* and *Ogataea polymorpha* yeasts. The protocol for the DNA content assay was also optimized to be suitable for both *Ogataea* and *Saccharomyces* yeasts. Using the HAQ approach, we demonstrate the expected effects on the cell cycle progression for several compounds and conditions and show usability in conjunction with additional fluorophores. Thus, our data provide a simple approach that can be utilized in a wide range of studies where the effects of various stimuli on the cell cycle need to be monitored directly in living cells.

## 1. Introduction

Yeasts are commonly used to study the eukaryotic cell cycle. Some yeasts, like various *Candida*, are pathogens, while others, like *Saccharomyces* and *Ogataea,* are important biotechnological platforms. In both basic and applied research involving yeast, detection of cell cycle perturbation is often highly relevant. Researchers have used a method that is well-validated in higher eukaryotes to obtain single-cell data on DNA content. This involves fixing cells with alcohols, removing proteins and RNA via enzyme treatment, and then staining them with DNA-specific dyes to assay whether cell populations display changes in their DNA content. Detection of DNA staining is most often accomplished by using flow cytometry, which is a quick, reliable, and quantitative method. However, preparation of cells for this assay is somewhat laborious, and so this approach is not commonly employed in high-throughput studies, with notable exceptions [1].

Histones have been studied intensively in model organisms and in humans. Their major function involves packaging DNA into the nucleus as chromatin, and they are also model proteins for the cell-cycle-dependent regulation of gene expression. However, the histone genes are atypical in that there are multiple copies of most histone genes in all organisms. In this regard, yeasts are a highly tractable model organisms, since, for instance, the budding yeast *Saccharomyces cerevisiae,* has only two copies of each of the major core histone genes [2]. Histones also extensively participate in gene regulation via both classical and epigenetic mechanisms (see [3] for a review). Notably, even a very small excess of histones over DNA is toxic [4] and can aggregate and precipitate chromatin. This problem occurs when DNA charge neutralization exceeds a critical value [5]. Thus, cells regulate the relative and absolute amounts of histones and their mode of delivery to the DNA very tightly [6] and histones are a notable group of proteins whose abundance does not depend on cell size directly, but rather on the amount of DNA. This is seemingly achieved at the promotor level by attracting a saturated amount of transcriptional machinery [7].

This suggested to us that GFP-tagged histones could be used as convenient tools to monitor the yeast cell cycle using flow cytometry. In this paper, we prove this by optimizing the classic protocol of standard DNA-staining and comparing it with GFP-tagged histone abundance quantification for two yeast genera in various conditions that induce changes to the cell cycle state of yeast populations.

## 2. Materials and Methods

### 2.1. Yeast Strains and Transformation

This study used *S. cerevisiae* strains from the collection of strains with GFP-tagged genes [8], the unmodified BY4741 strain [9], and *Ogataea parapolymorpha* DL1-L [10]. *O. parapolymorpha* was transformed according to the modified PEG-Li-acetate method [11].

### 2.2. Plasmid Construction, Obtaining HTB2-GFP Fusion in O. parapolymorpha, and R-GECO Expression Cassette Genome Integration

To construct the cassette for obtaining fusion of the *O. parapolymorpha HTB2* with the GFP-encoding sequence, the plasmid pAM779 [12], which consisted of the self-excising vector pAM773 [13] and the codon-optimized sequence encoding tagGFP2 [14,15], were used as a vector. The cassette was constructed by the insertion of inversed recombination arms into this vector according to the previously described scheme [16]. The inversed recombination arms directing the recombination with the *O. parapolymorpha HTB2* locus were obtained as follows. The *O. parapolymorpha* genomic DNA was digested with Ecl136II, self-ligated and used as a template for PCR with the primers CTCCTCACCTCCAGACATGTAAGCAGAGGTGGCAGAAGTG and ACGAAGTTATTAGGTGATGGTACGGTGTATTGGGTATAA, which possessed extensions complementary to the vector sequences. The obtained product was inserted between the PciI and EcoRV sites of the pAM779 vector by the ligation independent fusion cloning. Prior to the yeast transformation, the obtained plasmid designated as pAM984 was digested with Ecl136II to ensure double-crossing-over recombination with the genomic *HTB2* locus. Clones with the plasmid integrated into the target locus were revealed by PCR analysis according to the scheme depicted in the Appendix A. The vector sequence was self-excised via the procedure described previously [17].

The *O. polymorpha* codon-optimized R-GECO encoding sequence was synthesized by GenScript (Piscataway, NJ, USA). This sequence was placed under the control of the *OpaMAL1* promoter and inserted in a shuttle vector possessing G418 resistance and *LEU2* selectable markers. The obtained plasmid was designated as pKAM930. To avoid a complex description of the plasmid construction, its sequence and map is presented in the Appendix A. Prior to yeast transformation, the plasmid was digested with PdmI and Alw44I. This removed the *LEU2* selectable marker and facilitated plasmid integration into the genome. *O. parapolymorpha* transformants obtained from the G418-containing medium were replica plated onto a medium containing sucrose as a sole carbon source to induce the *MAL1* promoter. The R-GECO fluorescence was induced by illumination with a green (~525 nm) light emitting diode and viewed through a red filter. One of the clones with the brightest fluorescence was used to fuse its *HTB2* with a GFP-encoding sequence, as described above.

### 2.3. Yeast Cell Fixation, Staining with PI and SYBR Green and Flow Cytometry

According to the previously described procedure the cells from 0.4 mL culture were precipitated by 3 min centrifugation at 2000× *g* in a Eppendorf Minispin centrifuge (Ulm, Germany), washed with distillate water, and fixed by suspending in ice-cold 70% ethanol [18,19]. To achieve immediate fixation, 0.4 mL of the yeast culture was rapidly mixed with 1 mL ice-cold 96% ethanol in 1.5 mL microcentrifuge tubes. After a 15–20 min incubation, cells were precipitated by centrifugation. The pellet was thoroughly re-suspended in 1 mL of water by intense vortexing to ensure complete pellet disintegration. Cells were spun down again, re-suspended in 70% ethanol and stored in at 4 °C until the staining.

To perform the staining procedure, aliquots of suspensions of the fixed cells were centrifuged and cells were re-suspended in water to obtain OD600 0.5–0.6. Cell from 0.5 mL of the obtained suspensions were precipitated by centrifugation and re-suspended in 50 μL of TE buffer (10 mM Tris-HCl pH 8.0 1 mM EDTA) containing RNase (1 mg/mL) and a DNA specific dye, PI (1 μg/mL) or SYBR Green (1 μg/mL) [18]. The cell suspension was incubated overnight with agitation at room temperature. Prior to the flow cytometry assay, it was diluted 15-fold with water.

Suspensions were analyzed on a Cytoflex S flow cytometer (Beckman Coulter, Brea, CA, USA) equipped with a 96-well sampler. GFP and SYBR Green fluorescence was assayed using the488 nm laser and 525/40 FITC filter, whereas Propidium Iodide (PI) staining, as well as the R-GECO calcium sensor, were assayed with the 532 nm laser and 585/42 PE filter. Filters and lasers were supplied with the instrument. 

## 3. Results

### 3.1. Optimization of the Protocol for DNA Content Assay in Yeast Cells by PI and SYBR Green Staining

The main goal of this study was to establish whether assaying a fluorescent-protein-tagged histone in live cells can provide information on cell cycle progression similar to DNA staining with fluorescent dyes like PI or SYBR Green in fixed cells. We also aimed to test this approach in distantly related yeasts to show its versatility. Specifically, *S. cerevisiae*, *O. parapolymorpha* and *O. polymorpha* were chosen for this study. Although a number of different protocols for DNA content assay in fixed *S. cerevisiae* cells have been published, we could not find any for *O. parapolymorpha* and the closely related species *O. polymorpha*. The most noticeable difference between the *S. cerevisiae* protocols is the treatments of fixed cells with a protease. Different proteases, such as proteinase K or pepsin, can be used for this purpose [20], however, this step is sometimes omitted [21]. Also, such protocols start from chilling cells, washing with water, and subsequent ethanol fixation. It is possible that during chilling and washing steps, cells can proceed with the cell cycle in conditions differing from those defined by the experiment. So, to optimize the *S. cerevisiae* protocols for *Ogataea* yeasts, we studied whether cells can be rapidly fixed prior to the washing and whether protease treatment is actually required.

To perform rapid cell fixation, an aliquot of exponentially growing YPD culture was mixed with 2.5 volumes of cold ethanol (see Section 2). The cells spun down by centrifugation formed a rigid pellet, which was difficult to re-suspended in 70% ethanol. We supposed that some components of the medium precipitate with the cells and cause this adhesion. Indeed, washing the precipitated cells with water lead to dispersion of the pellet that allowed further manipulations.

Treatment of the *O. polymorpha* cells with proteinase K was found to be too harsh since it led to significant distortion of the cell shape and cell adhesion that made analysis by flow cytometry impossible. Treatment with pepsin did not cause such problems, However, we did not find any difference between treated and untreated cells (Appendix A). Thus, the final version of the modified protocol included rapid fixation by mixing cell culture with chilled ethanol and omitted the protease treatment step.

To test whether the rapid fixation step affects cell staining with the DNA binding dyes, PI and SYBR Green, *S. cerevisiae*, *O. parapolymorpha,* and *O. polymorpha* cultures in exponential and stationary phases in three replicates were subjected to the staining procedure with or without the rapid fixation step. According to the flow cytometry analysis (Figure 1), the rapid fixation step did not decrease the cell staining but made it more even since the obtained curves were noticeably better reproduced compared to those obtained without the rapid fixation. We do not have explanation of this effect, but it provides a noticeable advantage of applying the rapid fixation.

The exponentially growing culture contains cells at different stages of the cell cycle. Their distribution according to the DNA content exhibits two major peaks 1c and 2c, which correspond to cells that have not initiated DNA replication and have completed it, respectively. In *S. cerevisiae* we did not observe a noticeable redistribution between these peaks depending on the rapid fixation step. At the same time, in case of at least two replicates of the *O. polymorpha* cultures, the 1c peak became higher if the rapid fixation step is omitted, which could be expected if the washing and centrifugation steps prior to fixation allow some cells at the late stages of the cell cycle to proceed with division, while cells at the early stages of the cell cycle do not start DNA replication. We must note that definitive proof of this hypothesis requires additional study.

At the stationary phase, a yeast culture should not contain dividing cells due to the nutrient shortage, and one could expect to observe only a 1c peak. Indeed, the transition of the cultures from exponential growth to the stationary phase was accompanied with significant reduction or disappearance of the 2c peak. This was clearly observed independently of the fixation procedure. The obtained results demonstrate that the modified fixation-staining procedure is effective and applicable to these yeast species.

### 3.2. Comparison of GFP-Tagged Histone Abundance Quantification to DNA Content Measurement

In order to test whether GFP-tagged histones could provide information similar to DNA-content measurements, we first studied which histones are better suited to follow DNA content in yeast cells. To do this, we used *S. cerevisiae* strains from the systematic collection of the strains with GFP-tagged proteins [8]. Exponentially growing cultures of strains producing 10 different tagged histones were assayed by flow cytometry (Figure 2). In the case of the histones Hhf1, Hho1, Hta2, Htb1, Htb2, and Htz1, the GFP fluorescence significantly exceeded the autofluorescence (-GFP), However, only in cases of Hta2 and Htb2 did we observe pronounced bimodal distributions of fluorescence intensity, similar to the DNA content distributions (Figure 1), indicating a correspondence between histone amount and the DNA content. Based on these observations, Htb2-GFP was chosen for the consequent experiments because this fusion protein has been used more commonly in the literature for the in vivo observation of the nucleus. An *O. parapolymorpha* strain with GFP-tagged Htb2 was also constructed (see Section 2). 

The reasons for the differences between different histone proteins are unclear. One reason is defective strains from the GFP library, since at least the histone Hhf2 was previously reported [22] to exhibit bimodal distribution. This logic is likely applicable to the strains that exhibited fluorescence close to that of the control. Htz1 is known to be present only in some nucleosomes, thus its level might not follow DNA amounts precisely [23] and is lower than that of some of the core histones. The reasons for the differences between Htb1 and Htb2, as well as Hta1 and Hta2 are unclear, However, Huh et al. [8] also note the lower expression of levels of Htb1/Hta1 compared to Htb2/Hta2. Possibly, the presence of the GFP tag influences the abundance of the first pair, but not the second. Although in our data, Htb1 is relatively abundant, it does not display clear peak separation. Another possibility is that the GFP tag affects protein functionality more strongly for some histones compared to others and thus skews the results.

To directly compare whether and how well the Htb2 content correlates with DNA content under conditions of different cell culture density, samples of the same liquid cultures of *S. cerevisiae* and *O. polymorpha* strains expressing Htb2-GFP were directly analyzed by flow cytometry to assay Htb2-GFP abundance or underwent optimized fixation and SYBR Green staining procedure to assay DNA content. Note that the ethanol fixation step completely abrogates GFP fluorescence. Distributions of cell fluorescence according to the GFP and SYBR Green fluorescence were very similar indicating that the histone amount closely follows the DNA content in both yeast species independently of the growth phase (Figure 3a). A notable feature is the imperfect alignment of peaks obtained in different conditions and this varied somewhat between repeats. The reason for this is not clear as of now, however, in most cases it does not hamper interpretation of the data and it is observed both for DNA staining as well as histone abundance quantification (HAQ).

### 3.3. Effects of Cell Cycle Perturbations and Antifungal Drugs on DNA and Htb2 Histone Content Are the Same, Allowing the Use of HAQ for Rapid Assay of Effects on the Cell Cycle

To study whether assaying the Htb2 content allows monitoring cell cycle arrest and perturbations in response to compounds with known effects on the cell cycle, we used hydroxyurea and 5-fluoro cytosine, which are known to affect the cell cycle (Figure 3b). For these compounds, we observed the expected disappearance of 1c peak and shifting of most of the cells into the 2c peak according to the DNA staining as well as HAQ.

### 3.4. Use of HAQ in Conjunction with Another Fluorophore

To show that HAQ can be easily combined with other fluorophores, we created *O. parapolymorpha* strains expressing Htb2-GFP and the calcium sensor R-GECO, which exhibits red fluorescence and increased brightness at higher calcium concentrations [24]. A related green ratiometric calcium sensor, GEM-GECO, was recently tested in *O. parapolymorpha* by our group [25] and was shown to be functional. However, it was not suitable for our task since its fluorescence spectrum is highly similar to the GFP used to tag Htb2. That was why we used R-GECO to demonstrate the possibility of monitoring an additional fluorescent label. We could easily observe the baseline amounts of R-GECO in unperturbed cells expressing Htb2-GFP (Figure 4a), and could also, by normalizing the data to cell size (FSC), observe that the R-GECO fluorescence was somewhat higher in the 2c cell fraction (Figure 4a,b). A trivial reason for this might be increased expression of R-GECO in 2c stage cells, however, it is also possible that 2c cells exhibit higher amounts of calcium. The latter possibility is supported by previous reports of calcium pulses during cell division in *Schizosaccharomyces pombe* [26], However, a definitive conclusion requires additional experiments which are beyond the scope of this work.

## 4. Discussion

Because, based on the literature [7] and the functions of histones, it was reasonable to assume that DNA amount was correlated with histone amounts, we aimed to verify that GFP-based histone levels were correlated with the DNA content. At least two yeast histones (Hta2 and Htb2) were proven to be suitable, i.e., provided distributions of fluorescence intensity similar to DNA abundances. In order to see whether the data obtained via GFP-based histone abundance quantification was in accordance with DNA-staining under perturbed conditions, we assayed exponentially growing and stationary cultures of two yeast species, namely *S. cerevisiae* and *O. parapolymorpha*, whose *HTB2* genes were tagged with a GFP-encoding sequence. In all cases the histone content distribution pattern in living cells followed that of the DNA content in the fixed cells, including cell cultures at different stages of medium depletion, as well as treatment with compounds that are known to affect the cell cycle of yeast. This makes HAQ a potentially very useful approach for high-throughput screening of cell cycle perturbing conditions and compounds. The main advantages of HAQ are a) not needing any treatment of living cells prior to analysis, and b) the fact that HAQ can be easily combined with other fluorescent markers (such as compatible fluorescently labelled proteins or dyes) to determine how the cell cycle phase affects various cellular parameters, as shown in Figure 4.

Qualitatively, the two methods seem to be highly congruent, albeit we do not propose that HAQ replaces or should be used instead of DNA staining, since the latter is a direct assay of the DNA content, while the former may be affected by additional parameters, such as changes to translation rates, or other unknown factors. Highlighting this, the ratio between the presumed 1c and 2c fractions is similar, but not identical for the two methods (Table 1). For *S. cerevisiae*, Htb2 histone shows a lower ratio between the 1c and 2c peaks, while in *O. parapolymorpha*, it is the other way around. The reason for this is unclear. One possibility is that the histone level might follow the DNA content with some delay because the protein needs to be produced by cells. Our data also indicate the possibility that these phenomena can be species-specific, but this requires further investigation. However, despite these caveats, our results show that HAQ can be a useful approach for simple and high-throughput assay of cell cycle states of yeast cells.

## Figures and Tables

**Figure 1 jof-09-01033-f001:**
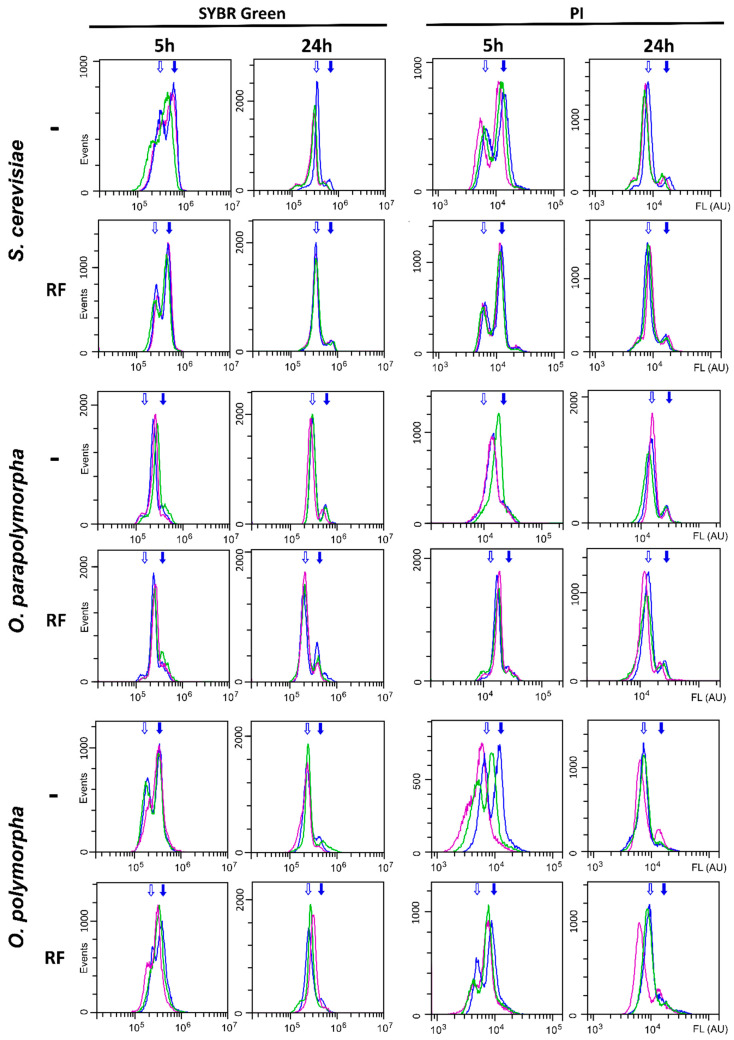
Distribution of *S. cerevisiae*, *O. parapolymorpha,* and *O. polymorpha* PI- (PI) or SYBR Green-stained cells according to their fluorescence intensity. Three independent overnight saturated YPD cultures were diluted 50-fold with fresh YPD and incubated at 37 °C (*Ogataea*) or 30 °C (*S. cerevisiae*) for 5 h to obtain rapidly dividing cultures or 24 h to obtain stationary cells. The cells were fixed and stained by the procedure involving (RF) or not (-) the rapid fixation step with cold ethanol (see Section 2). Diagrams representing independent replicates are shown in different colors. Empty arrows mark the peak of the 1c population, filled arrows—2c. For ease of viewing, only the positions of the blue peaks are marked.

**Figure 2 jof-09-01033-f002:**
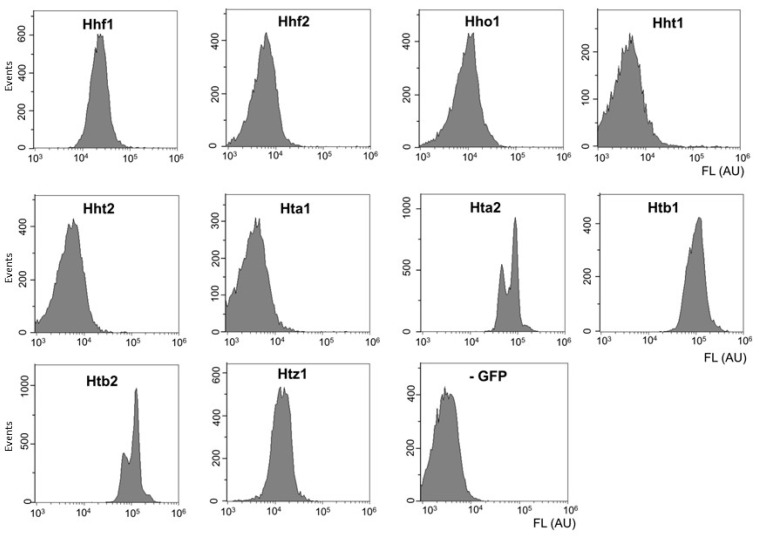
GFP fluorescence in individual cells of *S. cerevisiae* strains expressing GFP-tagged histone genes. The BY4741 strain, which does not express GFP, was used as an autofluorescence reference (-GFP). Strains were grown overnight in YPD medium at 30 °C, then diluted 50-fold and then grown for an additional 5 h.

**Figure 3 jof-09-01033-f003:**
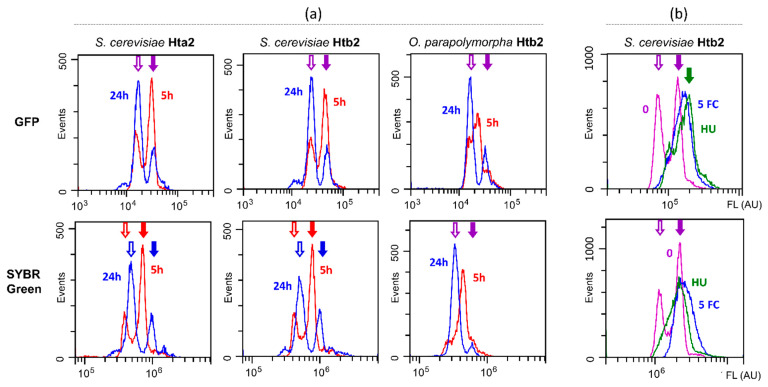
Flow cytometric analysis of yeast cells producing GFP-tagged Hta2 or Htb2 proteins according to DNA content (assayed by SYBR Green fluorescence) and GFP fluorescence. (**a**) Dependence on growth phase. Overnight saturated cultures were diluted 50-fold and incubated for 5 h to obtain logarithmic state cultures and for 24 h to obtain cultures in stationary phase. (**b**) Effects of 5-fluorocytosine (5 FC, 12.5 mcg/mL) and hydroxyurea (HU, 912.5 mcg/mL) compared to the untreated control (0). Logarithmic stage cultures were treated for 6 h and then collected for analysis. Empty arrows mark the peak of the 1c population, filled arrows—2c. Purple arrows depict the common location of peaks for distributions in (**a**), red/blue—individual peaks if the distributions are not aligned. Common peaks in (**b**) are also purple, while shifted ones are green.

**Figure 4 jof-09-01033-f004:**
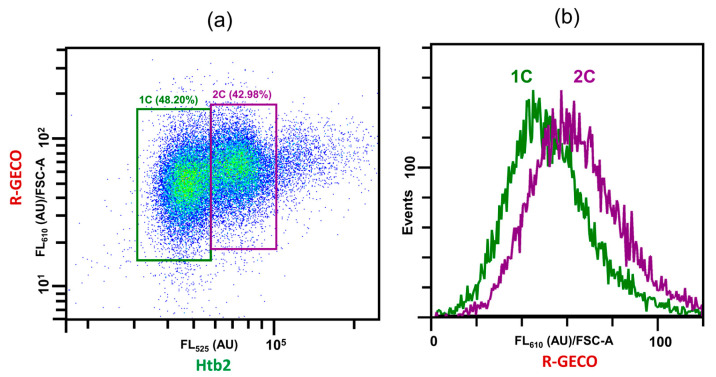
Htb2-GFP can be used in conjunction with a compatible red fluorophore for in vivo imaging. Flow cytometric analysis of *O. parapolymorpha* cells producing GFP-tagged Htb2 and the calcium sensor R-GECO. Cells expressing both proteins were grown overnight to saturated condition in YPD, 50-fold diluted with fresh YP with sucrose for induction of R-GECO, and incubated at 37 °C overnight, after which they were diluted 50-fold again and grown for an additional 5 h. The cells were then analyzed using flow cytometry. Gates based on the Htb2-GFP fluorescence in (**a**) were used to plot the 1c and 2c populations depicted in (**b**).

**Table 1 jof-09-01033-t001:** Ratios between the number of events (cells) in the 1c and 2c areas of the distribution of cells in logarithmic (5 h) and stationary phase (24 h) cultures of the *S. cerevisiae* Htb2-GFP and *O. parapolymorpha* Htb2-GFP strains from Figure 3 (images of the gate mappings are presented in Appendix A).

	*S. cerevisiae*		*O. parapolymorpha*	
Incubation Time	Htb2	DNA	Htb2	DNA
**5 h**	0.6	0.4	n/a *	n/a *
**24 h**	3.0	2.3	3.3	10.7

***** Calculations for the 5 h cultures of *O. parapolymorpha* were not performed due to the presence of a large number of cells with intermediate Htb2-GFP fluorescence (probably corresponding S-phase).

## Data Availability

Not applicable.

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
