# Peer review of "Histone Abundance Quantification via Flow Cytometry of Htb2-GFP Allows Easy Monitoring of Cell Cycle Perturbations in Living Yeast Cells, Comparable to Standard DNA Staining"

_jof, 2023, doi:10.3390/jof9101033_

Round 1

Reviewer 1 Report

This paper describes a new method for determining cell-cycle distribution in yeasts by measuring fluorescence of a GFP-tagged histone H2B gene. It is shown that the results obtained by this method is qualitatively similar to those obtained by traditional flow cytometry on fixed cells. It is somewhat surprising that nobody came up with this simple idea before. The experiments are nicely performed, and the paper deserves to be published. I only have two comments:

1) The quantitative difference between the two methods (e.g. Fig 3, middle panel) should be quantitfied by gating.

2) It should be briefly discussed why the different histone genes display different suitability to reflect DNA content. In addition, how this relates to the number of genes for each histone subunit.

Author Response

Reviewer 1 - This paper describes a new method for determining cell-cycle distribution in yeasts by measuring fluorescence of a GFP-tagged histone H2B gene. It is shown that the results obtained by this method is qualitatively similar to those obtained by traditional flow cytometry on fixed cells. It is somewhat surprising that nobody came up with this simple idea before. The experiments are nicely performed, and the paper deserves to be published. I only have two comments:

  • The quantitative difference between the two methods (e.g. Fig 3, middle panel) should be quantitfied by gating.

We have added quantitation via gating to Figure 3 (c), and the gates themselves are pictured in the Supplementary Figure 2.

2) It should be briefly discussed why the different histone genes display different suitability to reflect DNA content. In addition, how this relates to the number of genes for each histone subunit.

We have added the following paragraph to the manuscript -

The reasons for the differences between different histone proteins are unclear. One reason is defective strains from the GFP library, since at least the histone Hhf2 was previously reported [17] to exhibit bimodal distribution. This logic is likely applicable to the strains that exhibited fluorescence close to that of the control. Htz1 is known to be present only in some nucleosomes, thus its level might not follow DNA amounts precisely [18] and is lower than that of some of the core histones. The reasons for the differences between Htb1 and Htb2, as well as Hta1 and Hta2 are unclear, however [5] also note the lower expression of levels of Htb1/Hta1 compared to Htb2/Hta2. Possibly presence of the GFP tag influences the abundance of the first pair, but not the second. Although in our data, Htb1 is relatively abundant, it does not display clear peak separation. Another possibility is that the GFP tag affects protein functionality more strongly for some histones compared to others and thus skews the results.

**

We thank the reviewer for their time and their positive feedback about our manuscript

Reviewer 2 Report

The Authors claimed to establish a new method (by following the expression of GFP-tagged histone Htb2 using flow cytometry; HAQ) in yeast cells, which might replace the conventional DNA staining methods for monitoring of cell cycle. They intended to demonstrate that “rapid fixation” of the cells might provide more congruent data on the analysis of the cell cycle using the DNA staining methods. Unfortunately, the experiments described in the paper did not convince me for the following reasons:

1. According to the Authors the quanitative HAQ data did not match to that of obtained by the DNA staining method (Discussion, lanes 248-254). Although these data are missing, (therefore must be demonstrated), for me it means that the HAQ method is not suitable for replacing the conventional method to precisely monitor of the cell cycle.

2. The Authors mentioned that DNA-staining methods do not allow either distinguising the living and dead cells and or the simultaneus monitoring of other (fluorescent) cellular markers (Abstract, lanes 19-22). I did not find any experiments demonstrating that the HAQ method might overcome these problems.

Minor issues:

A. Introduction, lane 51-52. Appropriate references must be cited:

“In this regard, yeasts are a highly tractable model organisms, since, for instance, the budding yeast Saccharomyces cerevisiae has only two copies each of the major core histone genes [Ref]. Histones also extensively participate in gene regulation via both classical and epigenetic mechanisms [Ref]”.

B. Materials and Methods, lanes 69-72. Exact growth conditions must be described (e.g. growth media, temperature, cell number/mL during experiments).

C. Materials and Methods, lanes 73-88: How the O. parapolymorpha HTB2-GFP mutant was checked to rule out multiple/illegitime recombination events?

D. Results, lane 137-8. I haven’t found Figure S1: “The treatment with pepsin did not cause such problems, however we did not find any difference between treated and untreated cells (Figure S1).”

E. Figure 1.  The flow cytometry data of O. parapolymorpha PI and O. polymorpha SYBR Green staining should also be demonstrated.

F. Results, lane 162: Peaks 1c and 2c must be clearly indicated on the Figures.

Author Response

Reviewer 2

The Authors claimed to establish a new method (by following the expression of GFP-tagged histone Htb2 using flow cytometry; HAQ) in yeast cells, which might replace the conventional DNA staining methods for monitoring of cell cycle. They intended to demonstrate that “rapid fixation” of the cells might provide more congruent data on the analysis of the cell cycle using the DNA staining methods. Unfortunately, the experiments described in the paper did not convince me for the following reasons:

  1. According to the Authors the quanitative HAQ data did not match to that of obtained by the DNA staining method (Discussion, lanes 248-254). Although these data are missing, (therefore must be demonstrated), for me it means that the HAQ method is not suitable for replacing the conventional method to precisely monitor of the cell cycle.

We have included quantification for some for some of the data and addressed the raised concerns at the end of the discussion.

To clarify - we did not suggest that our approach can replace DNA staining, but that it can be used in many of the same circumstances, such as the study cell-cycle related phenomena. Because it is more easily applied in some situations, and can be used in vivo and in conjunction with other fluorophores, we think there are cases where it might be preferable. If one requires direct quantification of DNA for conclusions, this is possible to do that after obtaining HAQ data. 

  1. The Authors mentioned that DNA-staining methods do not allow either distinguising the living and dead cells and or the simultaneus monitoring of other (fluorescent) cellular markers (Abstract, lanes 19-22). I did not find any experiments demonstrating that the HAQ method might overcome these problems.

To address this concern, we performed additional experiments with a calcium sensor with red fluorescence (Figure 4).

The fact that HAQ works for additional fluorophores indicates that dead cell staining is possible, and we actually do have data to prove it (using propidium iodide), however they were obtained on several novel antifungal compounds that have not been characterized yet, and these data are included into a manuscript that is going to be submitted within a few months.

We have removed mentioning of dead/live cells from the manuscript to avoid making unsubstantiated claims.

Minor issues:

  1. Introduction, lane 51-52. Appropriate references must be cited:

“In this regard, yeasts are a highly tractable model organisms, since, for instance, the budding yeast Saccharomyces cerevisiae has only two copies each of the major core histone genes [Ref]. Histones also extensively participate in gene regulation via both classical and epigenetic mechanisms [Ref]”.

We have added these and additional references

  1. Materials and Methods, lanes 69-72. Exact growth conditions must be described (e.g. growth media, temperature, cell number/mL during experiments).

Growth conditions are described in the Figure legends, since they differed for different experiments

  1. Materials and Methods, lanes 73-88: How the O. parapolymorphaHTB2-GFP mutant was checked to rule out multiple/illegitime recombination events?

We have added information on the methods used to confirm correct integration (PCR amplification of the junction regions)

  1. Results, lane 137-8. I haven’t found Figure S1: “The treatment with pepsin did not cause such problems, however we did not find any difference between treated and untreated cells (Figure S1).”

The figure is included in the PDF file with the supplemental figures

  1. Figure 1.  The flow cytometry data of O. parapolymorpha PI andO. polymorpha SYBRGreen staining should also be demonstrated.

We have included the requested data in the modified version of Figure 1.

  1. Results, lane 162: Peaks 1c and 2c must be clearly indicated on the Figures.

We have labelled the 1c and 2c peaks on figures 1 and 3.

**

We thank the reviewer for their time and their useful comments which have, in our view, considerably improved the paper.

Round 2

Reviewer 2 Report

I'm sorry for the late answer: I have just allowed to leave the hospital...